# Regulation of Osteoclast Differentiation and Activity by Lipid Metabolism

**DOI:** 10.3390/cells10010089

**Published:** 2021-01-07

**Authors:** Haemin Kim, Brian Oh, Kyung-Hyun Park-Min

**Affiliations:** 1David Z. Rosensweig Genomics Research Center, Arthritis and Tissue Degeneration Program, Hospital for Special Surgery, New York, NY 10021, USA; kimha@hss.edu (H.K.); ohb@hss.edu (B.O.); 2Department of Medicine, Weill Cornell Medical College, New York, NY 10065, USA; 3BCMB Allied Program, Weill Cornell Graduate School of Medical Science, New York, NY 10065, USA

**Keywords:** osteoclasts, lipids, metabolism, fatty acids, cholesterol, statin

## Abstract

Bone is a dynamic tissue and is constantly being remodeled by bone cells. Metabolic reprogramming plays a critical role in the activation of these bone cells and skeletal metabolism, which fulfills the energy demand for bone remodeling. Among various metabolic pathways, the importance of lipid metabolism in bone cells has long been appreciated. More recent studies also establish the link between bone loss and lipid-altering conditions—such as atherosclerotic vascular disease, hyperlipidemia, and obesity—and uncover the detrimental effect of fat accumulation on skeletal homeostasis and increased risk of fracture. Targeting lipid metabolism with statin, a lipid-lowering drug, has been shown to improve bone density and quality in metabolic bone diseases. However, the molecular mechanisms of lipid-mediated regulation in osteoclasts are not completely understood. Thus, a better understanding of lipid metabolism in osteoclasts can be used to harness bone cell activity to treat pathological bone disorders. This review summarizes the recent developments of the contribution of lipid metabolism to the function and phenotype of osteoclasts.

## 1. Introduction

The skeleton is a dynamic tissue that undergoes constant remodeling by bone resorbing osteoclasts and bone forming osteoblasts [1,2]. In fact, 10% of the human skeleton undergoes remodeling every year [1]. Osteoclasts are derived from hematopoietic precursors and remove old and damaged bone [3,4,5,6,7]. Meanwhile, osteoblasts are derived from mesenchymal stem cells and form bone [8]. Bone-embedded osteocytes, derived from terminally differentiated osteoblasts, coordinate bone remodeling by detecting damage and sending a signal to osteoclasts and osteoblasts [9]. Bone remodeling is notably sensitive to environmental changes, including metabolic perturbation. Importantly, lipids in bone and bone marrow fuel this process and play an essential role in controlling the differentiation and function of bone cells [10].

Our understanding of lipid metabolism in bone has advanced significantly in the past decade. In addition to storing energy, lipids are an essential component of bone and provide cellular structure, including plasma membrane to cells. Lipids also transduce signals that are essential for cell survival and function, and are fused to other metabolic pathways in the cells. However, excessive lipid accumulation is directly correlated with changes in bone mass. Fat accumulation is inversely correlated with bone quality in obese humans, and high-fat-diet-fed mice show excess adiposity, low bone mineral density (BMD), and compromised bone quality [11]. In addition, increased fat mass in postmenopausal women has been shown to be correlated with decreased BMD [12]. Moreover, genetic defects in several genes involved in lipid metabolism are associated with changes in bone. However, the role and function of lipid metabolism in osteoclasts have not been well understood. This review summarizes the recent findings for the role of lipid metabolism in osteoclasts and discusses the effect of the regulation of lipid metabolism on osteoclasts and on the skeleton.

## 2. Lipid Metabolism

Lipids are composed of several species such as fatty acids, cholesterol, triglycerides (TGs), and phospholipids. The majority of lipids in bone are present within bone marrow, and a minority of lipids are in mineralized bone tissue [13]. Human bone marrow contains 28–84% of neutral lipids—including TGs, cholesterol, and free fatty acids—and less than 3% of phospholipids [14]. Lipid metabolic pathways process lipids and provide lipids to bone cells. They generate energy and are obtained by de novo lipid synthesis and the ingestion of dietary lipids. The dietary lipids are processed, and TGs and cholesterol are packaged into chylomicrons in intestinal epithelial cells. These complexes enter the lymphatic system and then into circulation; they subsequently acquire different apolipoproteins (Apo) including ApoB, ApoC-II, ApoC-III, and ApoE during the exogenous lipoprotein pathway. TGs and cholesterol are insoluble and are transported as a complex with other proteins. The TGs carried in chylomicrons are hydrolyzed in muscle and adipose tissue by lipoprotein lipase (LPL), which releases free fatty acids for cellular uptake and chylomicron remnants that are later taken up by the liver. Cholesterol is esterified into cholesteryl esters, packaged into lipid carrying lipoproteins, which are defined by their density. Packages of very-low-density lipoproteins (VLDLs) with TGs and cholesterol are synthesized in the liver and hydrolyzed in muscle and adipose tissue by LPL and VLDL remnants (intermediate lipoproteins, IDLs); free fatty acids are released for cellular uptake. VLDL remnants are further hydrolyzed by hepatic lipases to form low density lipoproteins (LDLs). LDLs bind to LDL receptors (LDLR) and are then taken up by tissues and cells. This endogenous lipoprotein pathway allows free cholesterol to be released into the numerous tissues and cells. Reverse cholesterol transport returns cholesterol to the liver and plays a key role in lipid homeostasis [15]. A cellular ABC transporter (ABCA1) mediates the first step of reverse cholesterol transport. Nascent high-density lipoproteins (HDLs) form mature HDL by acquiring cholesterol and phospholipids. Cholesterol efflux from cells to HDL is mediated by ABCA1, ATP binding cassette subfamily G (ABCG1), scavenger receptor B1 (SR-B1), or passive diffusion [16]. The HDL then transports the cholesterol to the liver via hepatic SR-B1 facilitated diffusion or by transferring the cholesterol to VLDL or LDL. Accumulation of cellular cholesterol leads to activation of several nuclear receptors, including liver X receptor α and β (LXRα and LXRβ), the retinoid X receptor (RXR), and the peroxisome proliferator-activated receptors (PPARα, PPARβ/δ, and PPARγ) [17]. These transcription factors also regulate the expression of cholesterol efflux transporters, including ABCA1 and ABCG1.

## 3. Regulation by SREBPs

Cellular lipid levels need to be tightly regulated, not only to meet metabolic demands but also to protect cells from excess lipid-mediated toxicity. The Brown and Goldstein group discovered the key mechanisms of endogenous lipid synthesis [18]. Sterol regulatory element binding proteins (SREBPs) and SREBP cleavage-activating proteins (SCAPs) are key factors that are required for endogenous lipid synthesis (Figure 1).

SREBPs are a family of transcription factors that bind to genes containing a sterol regulatory element (SRE) and control a panel of genes involved in intracellular lipid synthesis [19,20]. The three isoforms, SREBP1a, SREBP1c, and SREBP2, have different roles in lipid synthesis. SREBP1c is responsible for regulating genes for fatty acid synthesis, whereas SREBP2 primarily regulates cholesterol synthesis by controlling the expression of genes for cholesterol synthesis; SREBP1a participates in both the functions of SREBP1c and SREBP2, and overlapping functions among all three isoforms are also reported [21,22]. Importantly, the SCAP/SREBP system is regulated by cellular lipid levels. When sterol levels are sufficient, sterol is bound to SCAP in the endoplasmic reticulum (ER), and SREBPs and SCAP are tightly bound to Insig (insulin-induced gene) and remain inactive on the ER membrane [18]. When sterol levels are low, the SCAP and SREBP complex detaches from Insig and is transferred from the ER to the Golgi apparatus [23]. The nuclear domain of SREBP (N-SREBP), an active form of SREBP, is generated through the subsequent cleavage by site-1 protease (S1P) and site-2 protease (S2P) in the Golgi apparatus and is then translocated into the nucleus, leading to activation of the transcriptional program of target genes [24,25]. This SREBP-mediated feedback mechanism contributes to the tight regulation of lipid pathways in the body.

*Srebp2* mRNA was increased, whereas the *Srebp1* mRNA was unaffected in RANKL-induced primary osteoclast differentiation [26]. Inoue et al., 2015 utilized fatostatin, a SCAP inactivator, to inhibit SREBP activity and RANKL-induced bone loss by suppressing osteoclast differentiation [26]. Under physiological conditions, giving fatostatin did not change any of the bone parameters, but HDLs in serum were slightly reduced. In contrast, fatostatin treatment improved bone parameters, with fewer TRAP-positive osteoclasts per bone perimeter in the GST-RANKL-induced bone loss model. Jie et al., 2019 also showed that fatostatin suppressed osteoclastogenesis and NFATc1 expression. In addition, knock-down of SREBP2 using siRNAs largely suppressed both the number and size of osteoclasts, as well as NFATc1 expression, while overexpression of SREBP2 partially restored the inhibitory effect of fatostatin on osteoclast formation [27]. However, the function of SREBP2 in osteoclasts is still not clear, as fatostatin can suppress both SREBP1 and SREBP2 activation. However, evaluation of the direct role of SREBPs in osteoclasts using SREBP1/2-deficient mice has not been demonstrated yet.

## 4. Cholesterol Synthesis Pathway

Although more than 80% of daily cholesterol synthesis occurs in the liver and intestines, cholesterol synthesis (mevalonate pathway) also occurs in bone cells. This synthesis begins with acetyl-CoA derived from glucose, glutamine, or acetate metabolism [28] (Figure 2).

The sterol intermediates in cholesterol synthesis are named by the nomenclature system of the LIPID MAPS consortium (lipidmaps.org). These sterol intermediates have biological effects and are circulated in the bloodstream [29]. Cholesterol synthesis is strictly regulated by its demand; Schoenheimer et al., 1965 found that feeding mice cholesterol reduced its synthesis [30]. Among more than 20 enzymes in cholesterol synthesis, much attention has been focused on HMG-CoA reductase (HMGCR), an ER enzyme with its catalytic site facing the cytosol [31]. HMGCR is involved in the rate-limiting step in cholesterol synthesis and is regulated at multiple points [32]. HMGCR is a target of statins, a drug that lowers the concentration of cholesterol in the bloodstream. HMGCR is also a SREBP target but the transcriptional regulation by the SREBP pathway is slow. Additionally, HMGCR is rapidly regulated by post-translational mechanisms, including proteasomal degradation and AMPK-mediated S872 phosphorylation [33]. Targeting the components of the cholesterol biosynthesis pathway is an important way to regulate lipid metabolism.

## 5. The Role of Cholesterol in Osteoclasts

Cholesterol is an essential component of the cellular membrane and serves as a precursor for steroid hormones [34]. Cholesterol plays an important role in many cellular functions, and cholesterol biosynthetic pathways can interconnect with other signaling pathways [35]. Cholesterols constitute a significant portion of lipid rafts, which are membrane signal transducing platforms and play crucial roles in RANK-RANKL signal transduction during osteoclastogenesis [36,37]. In addition, phosphoinositide, a membrane lipid, regulates calcium signaling and osteoclast differentiation [38]. However, excess accumulation of cholesterol is highly deleterious to cells and underlies the pathogenesis of a number of metabolic diseases. High cholesterol levels also increase bone turnover. High fat diets in mice promoted osteoclastogenesis, which was followed by a decrease in bone mass [39]. The high-fat-fed antigen-induced arthritis (AIA) model also suggested that enhanced cathepsin K-positive osteoclasts contributed to more severe deterioration of the joints than in normal-diet-fed AIA rabbits [40]. Sanbe et al., 2007 showed that high-fat-fed rats experienced alveolar bone loss due to an increased number of osteoclasts; this was reversed by vitamin C supplementation [41].

Cellular cholesterol can be supplied by importing from lipoproteins or from cholesterol biosynthetic pathways. Macrophages share the same origin as osteoclasts. The low-density lipoprotein receptor (LDLR) on macrophages promotes the internalization of ApoB-containing lipoprotein, resulting in high levels of intracellular cholesterol. Increased cholesterol further subsequently downregulates LDLR expression in macrophages to inhibit the intake of cholesterol [42]. The regulation of cholesterol by lipoproteins also plays an important role in osteoclasts. Induction of cholesterol delivery by low density lipoprotein (LDL) significantly increased osteoclast viability, while the depletion of LDL suppressed osteoclast formation [43]. Osteoclast formation decreased when the cells were cultured in LDL-depleted serum, and resupplying oxidized LDLs reversed the impaired osteoclastogenesis in the LDL-depleted serum [44]. Cholesterol efflux from osteoclasts to high density lipoprotein (HDL) plays an important role in osteoclast apoptosis and fusion. Removing cholesterol through treatment with cyclodextrin or HDL induced osteoclast apoptosis by shutting off intracellular survival signals such as Akt, mTOR and S6K [45]. The depletion of cholesterol had been further found to suppress V-ATPase activity in osteoclasts by disturbing lipid rafts and affecting bone resorption [36]. Huang et al., 2018 showed that HDL3 promoted cholesterol efflux from osteoclasts by upregulating ABCG1, suppressing osteoclast fusion and survival [46]. Osteoclasts express both LDLR, a receptor of LDL, and SR-A, a receptor of modified LDL. LDLR-deficient mice had increased levels of circulating total cholesterol and LDL [47]. Tintut et al., 2004 showed that osteoclastogenesis was comparable between high-fat-fed control mice and high-fat-fed LDLR-deficient mice [48]. However, bone resorption activity and TRAP activity were significantly increased in osteoclasts from high-fat-fed LDLR-deficient mice. In contrast, two other studies reported that LDLR positively regulated osteoclastogenesis in control-chow-fed mice. LDLR-deficient cells exhibited reduced osteoclast differentiation and bone resorption, and LDLR-deficient osteoclasts also displayed fusion and survival defects [44]. LDLR knockout mice exhibited increased bone mass due to reduced numbers of osteoclasts [44]. This study additionally showed that impaired osteoclastogenesis in LDLR-deficient osteoclasts was rescued by the addition of cholesterol [45], suggesting that LDLR-mediated cholesterol uptake by macrophage might be required for osteoclastogenesis.

Although cholesterol treatment enhances RANKL-induced osteoclastogenesis, the underlying mechanism remains unclear. Sjogren et al., 2002 showed that increased IL-1α, an inflammatory cytokine, by cholesterol treatment contributed to enhanced osteoclastogenesis and bone resorption [49]. Wei et al., 2016 discovered cholesterol as an endogenous ligand for estrogen-related receptor alpha (ERRα) by utilizing affinity chromatography of tissue lipidome [50]. Cholesterol was found to exert a positive signal through ERRα to enhance osteoclastogenesis [50]. ERRα has been shown to be important for osteoclastogenesis [50,51,52]. While feeding WT mice a high cholesterol diet decreased bone mass, high cholesterol diets in ERRα KO mice had no effect on bone volume. In ERRα KO mice, bone volume increased in comparison with WT mice, but the treatment of zoledronic acid—an inhibitor of farnesyl diphosphate synthase—could not further enhance bone volume in ERRα KO mice. Thus, these studies demonstrated that cholesterols affect osteoclastogenesis via intrinsic and extrinsic activation.

Oxysterol, an oxidized form of cholesterol, is a ligand for LXR and suppresses SREBP2 activation [53]. Oxysterol has been identified as an EBI2 (Epstein-Barr virus-induced gene 2) agonist and is generated by cholesterol 25-hydroxylase (CH25H) [54]. EBI- or CH25H-deficient mice exhibited increased bone mass, and this phenotype protected female mice from ovariectomy-induced and age-induced bone loss [55]. Oxysterol and EBIs facilitated osteoclast precursors (OCPs) homing toward the bone surface and regulated bone homeostasis [55]. OCP homing was shown to be mediated by G⍺i protein-coupled receptor (GPCR) EBI2 and its oxysterol ligand 7a,25-dihydroxycholesterol (7a,25-OHC), abundantly secreted from osteoblasts [55]. OCPs also secreted 7a,25-OHC and further limited the migration of OCPs to bone surfaces by turning on auto-regulatory signals. Therefore, osteoclasts require cholesterol for their functions, although underlying mechanisms are incompletely characterized.

## 6. LXRs and RXRs

Liver X receptors (LXRs) are group 1 nuclear receptors that function as transcriptional regulators in lipid and cholesterol metabolism [56]. LXRα and LXRβ are cholesterol sensors and play an important role in cholesterol metabolism, inflammatory responses, and glucose metabolism [57]. LXRα specifically has shown functional importance as a cholesterol receptor in bone resorption activity and also regulates fatty acid metabolism by controlling the lipogenic program through SREBP1c [58]. Female LXRα knockout (KO) mice exhibited an increase in cortical bone parameters, although more endosteal osteoclasts were observed in LXRα KO mice. Serum levels of carboxy-terminal collagen crosslinks (CTX) and tartrate-resistant acid phosphatase (TRAP) were reduced in LXRα KO mice, indicating functional defectiveness of LXRα-deficient osteoclasts. LXR agonists positively regulated the genes related to lipogenesis and lipid transport, including SREBP1c and apolipoproteins [59]. These agonists also suppressed RANKL-induced c-FOS and NFATc1 expression and induced osteoclast apoptosis [60]. Further, activation of LXRs inhibited osteoclast differentiation by regulating AKT activation in a LXRβ-dependent manner [61]. LXRs form a heterodimeric complex with retinoid X receptors (RXRs) to bind to LXR response elements (LXRE) on target genes [62]. RXRs also control lipid metabolism. RXRα and RXRβ are expressed in osteoclasts. Loss of both RXRα and RXRβ in hematopoietic cells resulted in the generation of giant but resorption-defective osteoclasts [63]. Both cortical and trabecular bone mass increased in male RXRα/β KO mice, and the deficiency of RXRα/β protected mice from ovariectomy-induced bone loss. v-maf musculoaponeurotic fibrosarcoma oncogene family, protein B (MAFB) is a negative regulator of osteoclastogenesis [64]. RXR homodimer directly bound to the promoter of MAFB, and MAFB was transcriptionally regulated by RXRs in osteoclast precursor cells. In addition, activation of RXR and LXR heterodimers indirectly regulated the expression of MAFB through induction of SREBP-1c [64]. LXRs and RXRs contribute to osteoclast differentiation and activity.

## 7. Fatty Acid Synthesis and Fatty Acid Oxidation

Fatty acids (FAs) accumulate in the skeleton and play an important role in the maintenance of bone structure in mice [65]. FAs are generated by lipid metabolism and are taken up by bone cells. If fatty acid levels are low, cells generate lipid by fatty acid synthesis pathways (Figure 3A).

Malonyl CoA is generated from cytoplasmic acetyl CoA by acetyl CoA carboxylase (ACC), and malonyl CoA can inhibit the activity of carnitine palmitoyl transferases (CPTs) [66]. Through a series of condensations of malonyl CoA by multifunctional fatty acid synthase (FASN), palmitate is generated [67]. Key enzymes of de novo FA biosynthesis pathway are induced by SREBPs, MondoA, and ChREBP (carbohydrate-responsive element-binding protein) [68,69].

Fatty acid oxidation (FAO, β-oxidation) is the major pathway for the degradation of FAs and mainly occurs in mitochondria (Figure 3B) [70]. FAs are synthesized by utilizing ATP and subsequently converted into fatty acid acyl-CoA in the cytosol. FA-acyl-CoA then enters the mitochondrial matrix, is broken down to acetyl-CoA units, and generates NADH and FADH_2_ for every two carbon units released. More specifically, fatty acyl CoAs are conjugated to yield carnitine by CPT1A, a mitochondrial outer membrane enzyme, and are transported into the mitochondrial matrix. In the mitochondria, CPT2 removes carnitine, and beta oxidation is controlled by a series of enzyme-mediated reactions shown in Figure 3B. Acetyl-CoA, NADH, and FADH_2_ are produced during the FAO process and fuel several metabolic pathways. FAs are stored for energy storage, and FAO provides and maintains energy homeostasis.

## 8. Function of Fatty Acid in Osteoclasts

FAs contain a straight alkyl chain—of which the number of carbons in the chain varies—and consist of a hydrophilic carboxylate group bound to a hydrophobic hydrocarbon chain. The classification of FAs is based on the number and position of double bonds within the hydrocarbon side chain. FAs are largely classified into two groups: saturated FAs containing no double bonds and unsaturated Fas with one or more double bonds. Fas are further categorized as long-chain fatty acids, which include polyunsaturated fatty acids (LCPUFAs), monounsaturated fatty acids (LCMUFAs), saturated fatty acids (LCSFAs), medium-/short-chain fatty acids (MCFAs/SCFAs) as well as their metabolites. These Fas are closely associated with bone health as well as bone disorders [71].

Long chain polyunsaturated fatty acids (LCPUFAs) are fatty acids with a minimum of 18 carbons and 2 double bonds. The simplest form of LCPUFAs is linoleic acid (18:2*n*-6; a precursor of ω6 PUFAs) and α-linolenic acid (18:3*n*-3; a precursor of ω3 PUFAs), which are essential fatty acids. Ω-3 LCPUFAs include eicosapentaenoic acid (EPA) and docosahexaenoic acid (DHA), and ω-6 LCPUFAs include arachidonic acid (AA) and γ-linolenic acid (GLA). LCPUFAs are mainly found in dietary sources. The main sources of EPA and DHA are fatty fish and sea food, and alpha-linolenic acid (ALA) is primarily found in plant oil [72]. In humans, ALA can be also metabolized to EPA and then converted into DHA [73,74]. DHA can be converted into EPA [75]. LCPUFAs are required to maintain the adequate concentration of DHA in the membrane and to be a precursor of prostaglandin, and play an important role in the lipoprotein metabolism [76]. However, the capacity of the metabolic process from ALA to EPA and DHA is relatively low, and thus dietary uptake of LCPUFAs is important [77]. LCPUFAs can directly suppress osteoclastogenesis and osteoclast activity [78,79]. LCPUFAs, including AA and DHA, suppressed osteoclast formation from human CD14-positive monocytes, resulting in reduced bone resorption activity [78]. This negative effect of DHA on osteoclastogenesis was mediated by the reduction of key signaling transductions pathways, including reduction of the activation of JNK, ERK, p38 MAPK, and NF-ĸB [80]. Another study also showed that both ω-3 and ω-6 LCPUFAs suppressed osteoclastogenesis in Raw264.7 [81]. These studies suggest that LCPUFAs negatively regulate osteoclast differentiation. Consistently, several other studies demonstrated the protective effect of the supplementation of LCPUFAs on pathological bone destruction. Intake of ω-3 LCPUFAs was positively associated with BMD in humans [82,83]. In addition, inflammatory bone loss induced by lipopolysaccharide-induced inflammation in vivo was protected by DHA supplementation [84]. Supplementation of ω-3 LCPUFAs reduced bone resorption in rats with pulp exposure-induced apical periodontitis and enhanced bone formation in the periapical area [85]. Fong et al., 2012 also showed that the maternal dietary delivery of ω-3 LCPUFAs to offspring increased bone formation while decreasing bone resorption in male offspring at a young age [86]. While the effect of LCPUFAs on osteoclasts remains consistent, more clinical prospective studies will be needed to establish the long-term effect of LCPUFA supplementations on bone health [77]. LCPUFAs bind to GPR40 and GPR120; a synthetic agonist of GPR 40/120 can mimic the inhibitory effects of fatty acids on osteoclastogenesis [87]. GPR40 knockout mice have been found to exhibit reduced bone mass. GPR40 and its downstream signaling mediate the inhibitory effect on osteoclast differentiation [88]. Another study also showed that GPR 120 activation blocks RANKL-induced osteoclast differentiation [89]. GW9508, a GPR40 agonist, induced necrosis-associated cell death in osteoclast precursor cells due to mitochondrial oxidative stress, suggesting that one of the mechanisms of osteoclastogenesis inhibition is free fatty acid receptor signaling [90]. Leukotriene B4, which can be converted from arachidonic acid by the enzyme 5-lipoxygenase, has been shown to stimulate osteoclastic bone resorption [91]. Either delivering leukotriene B4 over the mice calvariae or ex vivo organ culture of neonatal mouse calvariae increased osteoclast numbers and bone resorption. Unsaturated fatty acids (UFAs) have been shown to inhibit osteoclastogenesis from human CD14-positive monocytes by activating peroxisome proliferator activated receptors (PPARs) through inhibition of RANKL signaling [92]. PPARs are a family of nuclear receptors, and both monoUFAs (MUFAs) and polyUFAs (PUFAs) are natural ligands of PPARs [17]. While PUFAs increase PPARα, MUFAs activate PPARβ/δ. All PPAR activators suppressed osteoclastogenesis [92]. Accordingly, UFAs have a protective effect on bone [93].

Saturated fatty acids (SFAs) include apric acid (LA, C12:0), myristic acid (MA, C14:0), palmitic acid (PA, C16:0), and stearic acid (SA, C18:0) and have no double bonds between carbon molecules. SFAs can be synthesized de novo or be ingested. Monosaturated FAs, including oleic acid and palmitoleic acid, have one double bond. The effect of SFAs on osteoclasts is controversial, though most of studies report the beneficial effect of SFAs on osteoclast differentiation and activity. Oh et al., 2010 showed that SFAs prevented osteoclast apoptosis in a TLR4-dependent manner and that exposure to SFAs at late stages of osteoclast differentiation increased the formation of osteoclasts [94]. Among different SFAs, lauric acid (C12:0) and palmitic acid (C16:0) increase osteoclast survival by enhancing the production of macrophage inflammatory protein-1 alpha (MIP-1a) via activation of toll-like receptor 4/nuclear factor-kB signaling [94]. Drosatos-Tampakaki et al., 2014 showed that palmitic acids enhance RANKL-induced osteoclast differentiation and could induce osteoclast differentiation in the absence of RANKL [95]. Further, feeding a high-fat palmitic acid-enriched diet to mice accelerated bone loss compared to mice on a high-fat oleic acid-enriched diet; in contrast to palmitic acid, oleic acid (C18:1) is unable to activate osteoclastogenesis [95]. Increased accumulation of intracellular TGs is observed upon oleic acid treatment, and TGs suppress osteoclast differentiation. Diacylglycerol acyl transferase 1 (DGAT1) is an enzyme involved in triglyceride synthesis. DGAT1 knockout mice have larger osteoclasts and exhibit diminished bone mass due to the increased numbers of osteoclasts, and DGAT activation can suppress TNFα-mediated SFA-induced osteoclastogenesis. DGAT1 or oleic acid supplementation may have a protective role in bone loss. Cornish et al., 2008 showed that SFAs suppressed osteoclastogenesis [87]. Palmitoleic acid also inhibits RANKL-induced osteoclast differentiation from Raw264.7 through suppression of MAPK and NFkB signaling [96].

Short chain fatty acids (SCFAs) are the main metabolites that are produced in the gut from microbial fermentation of dietary fiber [97]. SCFAs regulate local and systemic immune functions [98]. Lucas et al., 2018 showed that SCFAs suppressed in vitro and in vivo osteoclastogenesis [99]. Feeding SCFAs to ovariectomized mice, collagen-induced arthritic mice, and K/BxN serum-transfer induced arthritic mice protected mice from bone loss [99]. The treatment of osteoclast precursors with SCFA shifted the metabolism toward glycolysis at early time points, which delayed the osteoclast differentiation process. The supplementation of the right probiotics, SCFAs, or diets that increase the endogenous production of SCFAs may assist in balancing osteoclast-mediated bone resorption and bone formation. Medium chain fatty acids (MCFAs), such as capric acid, also suppress osteoclastogenesis [100]. Different fatty acid species show the differential effect on osteoclastogenesis. As diets and microbiomes greatly contribute to the supply and processing of fatty acids, modulating FA metabolism may be linked to changes in bone.

## 9. Dyslipidemia and Bone Metabolism

Dyslipidemia is a risk factor for atherosclerosis, diabetes, cancer, and vascular calcification [101,102,103]. Recent studies also tie dyslipidemia to bone loss [48,104,105]. It has been suggested that hyperlipidemia may increase the risk of bone loss via regulating osteoclastic bone resorption and osteoblastic bone formation. Hypercholesterolemia induces bone loss [106]. Zhou et al., 2019 reported that males with hypercholesterolemia had a higher bone turnover rate and reduced BMD and that serum markers for bone resorption (CTX) and bone formation (P1NP) were negatively correlated with serum cholesterol [106]. A high cholesterol diet or increased endogenous cholesterol using ApoE-deficient mice results in bone loss in mice [39,106].

Dyslipidemia is also associated with postmenopausal status [12,107]. It has been reported that a decrease in BMD during menopause is accompanied by an increase in adipocytes in the bone marrow space [12,107]. Estrogen has been shown to play an important role in lipid regulation. Syed et al., 2008 showed that both estrogen and PTH treatment could reduce marrow adipocyte size in women with osteoporosis [108]. Estrogen signals via ERα regulate lipid droplet size and total lipid accumulation in the bone marrow space in vivo [109]. Ovariectomy in mice can also lead to accumulation of fat in the bone marrow [105]. Estrogen receptor alpha (ERα) knockout (ERαKO) mice exhibit increased lipid accumulation in bone marrow compared to wild-type mice or ERβKO mice. Accordingly, 17β-estradiol (E2) potentiates lipolysis, leading to fewer lipid droplets per cell in adipocytes while cells from ERαKO mice promote lipogenesis. The inhibitory role of estrogen in osteoclastogenesis may be, in part, mediated by suppressing lipid accumulation.

Obesity, a rising worldwide health problem, is also related to typical dyslipidemia, including increased TGs and free FAs, normal and increased LDLs, and decreased HDLs [110]. High body weight or BMI show a correlation with high bone mass [111,112], and thus obesity is considered a protective factor for osteoporosis by improving bone mass and maintaining high levels of estrogen. However, recent data reveals that obesity may be a risk factor for osteoporosis and fracture [113]. Moreover, accumulating evidence suggested that physical exercise is beneficial to bone health [114,115]. Physical inactivity and ageing are associated with obesity and osteoporosis [116]. The relative contribution of fat mass and lean mass on bone mass is controversial. Khosla et al., 1996 reported that both fat mass and lean mass are important for bone health [117]. However, recent reports showed that increased lean mass is also correlated with a higher BMD in pre- and postmenopausal women [118,119], suggesting that lean mass, not fat mass, is strongly associated with overall bone quality. Obesity is further associated with low-grade chronic inflammatory pathologies and leads to the development of co-morbid conditions, such as type II diabetes mellitus (T2DM). The metabolic changes associated with type I and II diabetes are accompanied by abnormal BMD [120]. Several studies have demonstrated that T2DM patients have normal or high BMD compared to age-matched healthy individuals [121]. Twenty-four men with T2DM were subjected for carotid intimal-medial thickness (CIMT)—an early diagnostic marker of cardiovascular events (e.g., atherosclerosis)—and BMD (femoral neck and lumbar spine) measurements [122]. The study indicated a negative correlation between femoral BMD and CIMT, suggesting a close relationship between atherosclerosis and osteoporosis in men with T2DM. However, there were no significant differences between groups regarding age, duration of T2DM, BMI, AC, SBP, DBP, statin use, smoking, HbA1C, cholesterol, or TGs. Poor bone quality of patients with diabetes is also associated with an increased relevant risk of hip fracture [123,124,125,126,127]. However, the difference in the risk of overall fractures of diabetes patients showed mixed results.

Dyslipidemia also affects conditions involving inflammatory bone loss such as periodontitis [128]. Chronic periodontitis often leads to alveolar bone loss due to an imbalance between active osteoclasts and inactive osteoblasts [129]. Bone-marrow-derived osteoclasts from db/db mice, a model of type-2-diabetes, are higher than control cells, but osteoclastogenesis is comparable between peripheral blood osteoclast precursor cells of patients with type-2-diabetes and those of control cells [130]. Although many studies have shown that dyslipidemic conditions lower BMD and lead to fragile bone quality, the effect of dyslipidemia on osteoclast differentiation and activity is incompletely determined.

## 10. Statins

Statins are FDA-approved drugs to treat patients with hypercholesterolemia and target HMG-CoA reductase, a key cholesterol synthesis enzyme [131,132]. Statins include simvastatin, lovastatin, atorvastatin, pravastatin, and mevastatin, and are the most prescribed cholesterol-lowering drugs. Statins have been shown to increase bone mineral density and to lower the risk of fracture [133,134,135] (reviewed in [136,137]). The positive outcome of statins on bone health is mostly due to the effect of lowering lipid levels [138].

### 10.1. Preclinical Studies

Statins directly affected bone cells, suppressing osteoclastogenesis and promoting osteogenesis [139]. Statins promoted osteogenic activity through the activation of the expression of bone morphogenetic protein-2 (BMP-2) and osteocalcin in mice [139], by suppressing farnesylation and/or geranylgeranylation of Rho/Ras small G proteins in osteoblasts [140]. Further, atorvastatin treatment in human osteoblasts increased osteoprotegerin (OPG), which antagonizes RANKL [141]. Statins also inhibited osteoclast activity [142]. Nakashima et al., 2013 showed that simvastatin treatment reduced osteoclast differentiation as well as interferon regulatory factor-4 (IRF4) expression [143]. Simvastatin inhibited DNA binding of IRF4 upon activation of Rho family of GTPases during osteoclast differentiation [143]. Administration of statins into rodents also increased bone volume [139]. Consistently, in vivo treatment of simvastatin prevented bone loss from an acute RANKL-induced bone-loss model [143]. Moreover, ovariectomized rats treated with simvastatin (20 mg/kg, twice daily for 3 months) showed increased bone mass, followed by decreased levels of serum TRAP-5b [144]. Lovastatin is another inhibitor for HMG-CoA reductase [145]. Dose-dependent inhibition of osteoclastogenesis by lovastatin has been shown in rat bone marrow cell cultures [146]. Overall, most of preclinical studies demonstrated that statins suppress osteoclastogenesis and osteoblast apoptosis, and promote osteogenesis.

### 10.2. Clinical Studies

Targeting hyperlipidemic conditions by statin in postmenopausal women has also shown to affect BMD. Statin treatment in 41 postmenopausal women increased bone mineral density (BMD) at the spine and hip compared to 100 control postmenopausal women [147]. Statin treatment also increased BMD in hypercholesterolemic postmenopausal women [148]. A clinical study that was conducted in 2006–2008 for 212 hyperlipidemia patients with osteopenia demonstrated that simvastatin, in comparison with other lipid lowering drugs (gemfibrozil or fibrate), significantly improved bone mass and other bone parameters [149]. Fifty-five postmenopausal women with T2DM were treated with lovastatin, and the lovastatin treatment group showed a clear improvement in BMD at lumbar spine and Ward’s triangle [150]. Statin also increase bone mass in T2DM patients. In the study with 37 women with type 2 diabetes mellitus (T2DM) who were taking simvastatin, there was an increase in the BMD of the simvastatin group, albeit not statistically significant [151]. Accordingly, 25-hydroxyvitamin D and 1,25-dihydroxyvitamin D were significantly increased in 91 hyperlipidemic patients who were treated with rosuvastatin for 8 weeks [152]. Another study included 122 patients with T2DM. After 2 additional years of follow up, it was evident that the use of HMG-CoA reductase inhibitor improved BMD in patients with T2DM [153]. Another study included 69 T2DM patients; 36 of whom were taking lovastatin, pravastatin, or simvastatin, and 33 patients in a non-treatment group. Significant higher percentages of BMD in the femoral neck, wards triangles, trochanter, and total hip had been observed in the statin group [154]. Statin treatment increased bone turnover markers in T2DM patients. Rosuvastatin increased osteocalcin compared to ezetimibe, a non-statin lipid lowering drug, in T2DM patients with hypercholesterolemia [155]. Statins also improved chronic periodontitis, and statin attenuated inflammatory bone loss of periodontitis [156,157,158]. Although some reports showed that statins have no effect on BMD or the risk of fracture [159,160,161], many studies reported the protective effect of statins on bone health (Table 1).

## 11. Conclusions

Controlling cellular and systemic lipid levels is essential for physiological bone homeostasis. Recent studies provide evidence that metabolic reprogramming is an important player for osteoclast differentiation and function [162]. Metabolic alteration has been identified in many bone diseases, including osteoporosis, and the close correlation between dysregulated lipid metabolism and bone has been increasingly appreciated. Accordingly, metabolism can be an attractive target of treatments for pathological bone resorption. Moreover, lipids can impact the phenotype of osteoclasts and osteoblasts in pathological conditions. While the role of lipids on osteoblasts and their bone formation activity are relatively well-studied, the molecular mechanisms of how lipid metabolism regulates osteoclast differentiation and activity are not yet fully elucidated. Thus, it is important to understand lipid metabolism in the context of osteoclastic bone resorption in dyslipidemic conditions. To understand the role of lipid metabolism in osteoclasts, several factors need to be considered. First, dyslipidemia can not only cause an increased lipid load in osteoclasts but also can indirectly affect osteoclasts by stimulating other cell types or by inducing inflammatory mediators. Thus, the role of crosstalk between osteoclasts and other cells in bone metabolism under dyslipidemic conditions needs to be further characterized. In addition to importing lipids from outside of cells, de novo lipid synthesis is active in osteoclasts. However, the differential effects, between exogenous lipids and endogenous lipids that are synthesized in osteoclasts, on osteoclast differentiation and function remain unclear. It is important to identify how the regulation of de novo lipid synthesis impacts osteoclast differentiation and function using in vivo mouse models, including mice that are deficient or overexpress key regulators of lipid metabolism, or fat-modifying-diet-fed mice (high-fat-diet-fed model, or low-fat/fat-free-diet-fed model). Thirdly, genetic variation in components of the transcriptional program of SREBPs such as LDLR, PCSK9, and HMGCR has been associated with lipid traits and a craniofacial phenotype [163,164]. It is important to identify whether the genetic variation in the genes related to lipid reprogramming also controls bone and the activity of osteoclasts. Lastly, lipid metabolism is closely linked to the different metabolic pathways, including glucose, glutamine, and acetate metabolism [162]. How the metabolic circuit between lipid metabolism and different metabolic pathways affects osteoclasts needs to be determined. Moreover, various metabolism-targeting drugs have been developed, and Enasidenib has been FDA-approved for treating any cancer [165]. However, the efficacy of these metabolic drugs on bone loss has not been fully evaluated yet. A better understanding of how lipid metabolism rewinds cellular energy to fulfill the energy demand for osteoclast differentiation and activity—especially in pathological conditions—will provide a new therapeutic insight into pathological bone loss.

## Figures and Tables

**Figure 1 cells-10-00089-f001:**
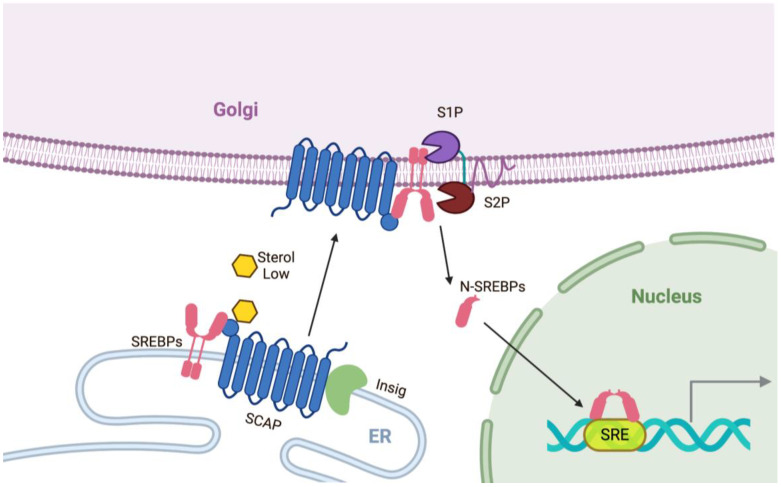
The regulation of sterol regulatory element-binding protein (SREBP, pink). SCAP (SREBP-cleavage-activating protein, blue) functions as a sensor for sterol. In the presence of high lipid/cholesterol/oxycholesterol, Insigs (Insulin-gene induced proteins, light green) are tightly attached to a SCAP and SREBP complex that remains in the endoplasmic reticulum (ER). Under low lipid/cholesterol/oxycholesterol conditions, Insigs are ubiquitinated and degraded, leading to the release of the SCAP and SREBP complex. The SCAP and SREBP complex is transported from the ER to the Golgi apparatus and is sequentially processed by cleavage by site-1 protease (S1P, purple) and site-2 protease (S2P, brown). The nuclear domain of SREBPs (N-SREBPs) are moved to the nucleus. Homodimers of N-SREBPs bind to SREs (sterol regulatory elements) in the promoter regions of genes encoding lipid pathway genes and other target genes.

**Figure 2 cells-10-00089-f002:**
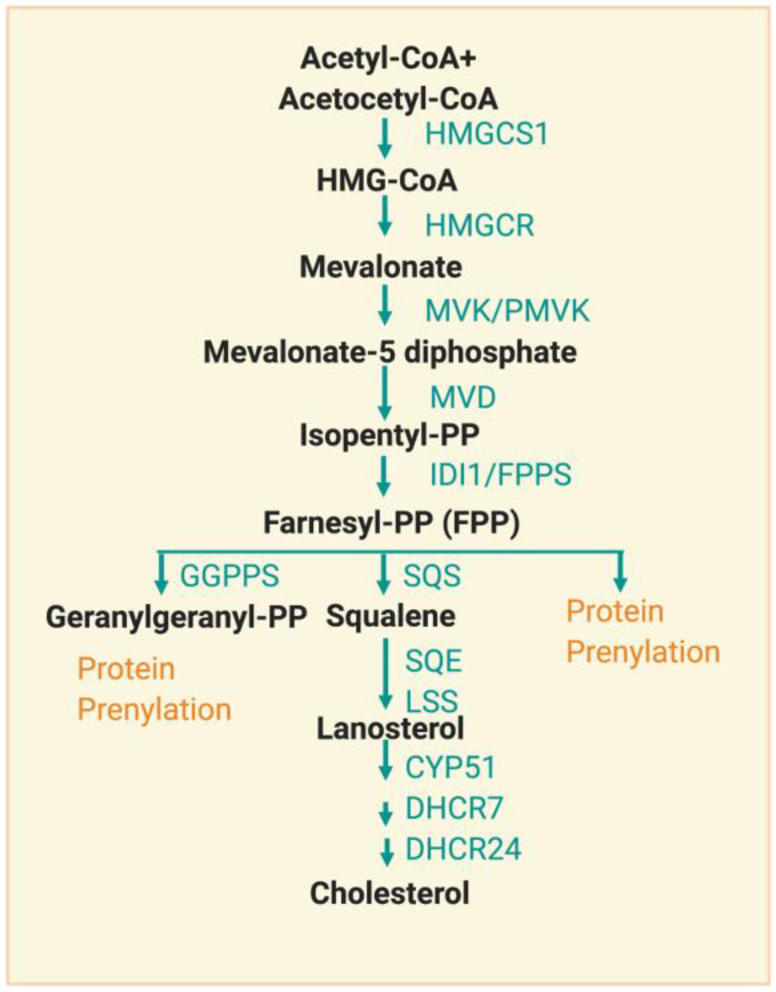
Overview of the cholesterol synthesis pathway. The cholesterol synthesis pathway uses acetyl-CoA and generates isoprenoid intermediates and cholesterol. Sequential molecules and key enzymes are described in a schematic diagram. HMGCS1, 3-hydroxy-3-methylglutaryl-CoA synthase 1; HMGCR, 3-hydroxy-3-methylglutaryl-CoA reductase; MVK, mevalonate Kinase; PMVK, phosphomevalonate kinase; MVD, mevalonate diphosphate decarboxylase; IDI1, isopentenyl-diphosphate delta isomerase 1; FPPS, farnesyl pyrophosphate synthase; GGPPS, geranylgeranyl diphosphate synthase; SQS, squalene synthase; SQE, squalene epoxidase; LSS, lanosterol synthase; CYP51, lanosterol 14alpha-demethylase; DHCR7, 7-dehydrocholesterol reductase; DHCR24, 24-dehydrocholesterol reductase. Green letters indicate enzymes and orange letters indicate protein modification.

**Figure 3 cells-10-00089-f003:**
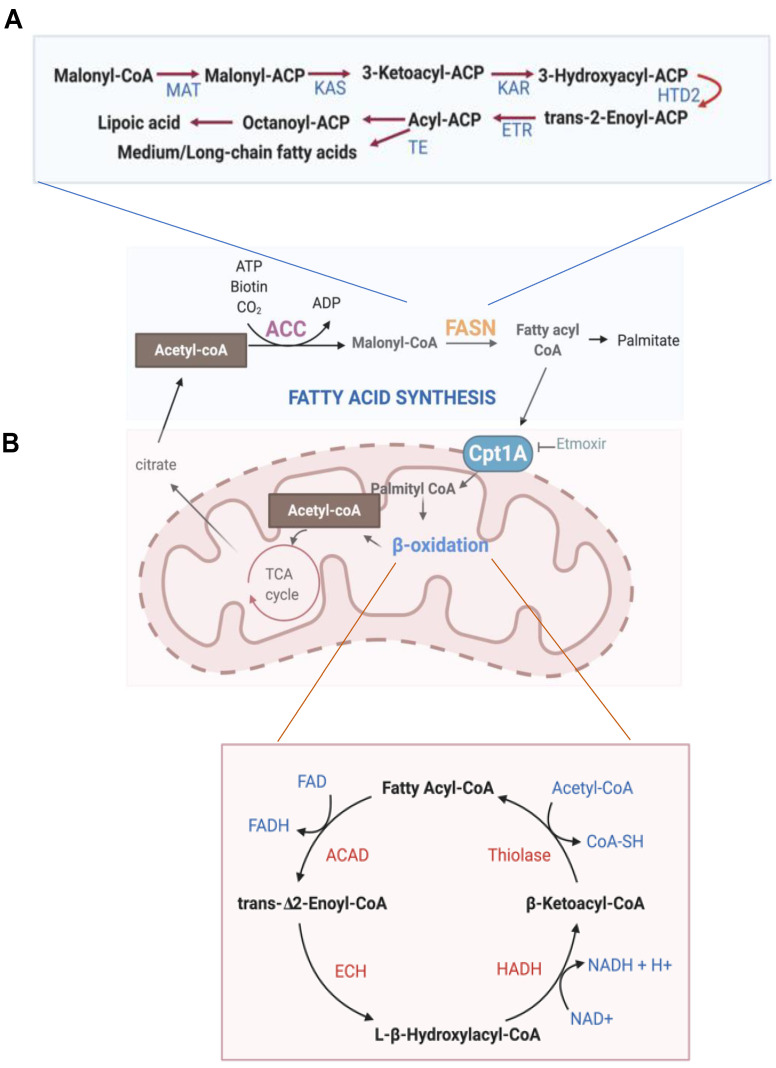
Overview of fatty acid metabolism. (**A**) Fatty acid synthesis pathway. Upper panel: the processing of malonyl-CoA by multifunctional domains of FASN. Sequential molecules and key enzymes are described in a schematic diagram. ACC, acetyl-CoA carboxylase; FASN, fatty acid synthase; FASN domains: MAT, malonyl-CoA transferase; KAS, ketoacyl synthase; KAR, ketoacyl reductase; HTD2, hydroxyacyl-thioester reductase type 2; ETR, enoyl-thioester reductase; TE, thioesterase. (**B**) Fatty acid oxidation pathway. Cpt1A (Carnitine Palmitoyl transferase 1A) catalyzes the rate-limiting step in fatty acid oxidation. Sequential molecules and key enzymes are described in the bottom schematic diagram. ACAD, acyl-CoA dehydrogenase; ECH, enoyl-CoA hydratase; HADH, hydroxyacyl-CoA dehydrogenase.

**Table 1 cells-10-00089-t001:** Statins in clinical studies.

Study Subjects	Compound	Group Size (# of Treatment)	Effects	Duration of Supplementation	Ref.
Elderly women	Statins	3028 (803)	lower cortical porosity and higher cortical bone density	Not reported	[133]
Healthy periodontium, chronic periodontitis + T2DM, chronic periodontitis, chronic periodontitis + T2DM+simvastatin	Simvastatin	80 (20)	Improved clinical parameters of periodontitis (also lowers IL-6 & TNFa).	5 to 10 years	[156]
T2DM with chronic periodontitis	Atorvastatin (Subgingivally local delivery)	75	Improved in reducing chronic periodontitis.	3, 6 and 9 months	[157]
T2DM with chronic periodontitis	Rosuvastatin (RSV) or atorvastatin (ATV) (Subgingivally local delivery)	90 (38)	Improved clinical parameters of periodontitis in RSV group relative to ATV and placebo groups	6 and 9 months	[158]
Hyperlipidemia with osteopenia	Simvastatin gemfibrozil fibrate	212 (106)	Improved bone mass and significantly higher bone turnover markers	18 months	[149]
Patients with T2DM and hypercholesterolemia	Rosuvastatin or ezetimibe	36 (18)	Increased serum osteocalcin	3 months	[155]
Postmenopausal women with T2DM	Lovastatin	55 (28)	Increased BMD in Lumbar spine and Ward’s triangle	18 months	[150]
Women with T2DM	Simvastatin	111 (37)	* Improved BMD-no significant changes	Mean duration 46 months	[151]
US veterans populations	Statins	91,052 (28,063)	Significant reduction of fracture risk	Not specified, but excluded from the statin group if individuals stopped statin for more than 12 months	[134]
Hypercholesterolemic postmenopausal women	Simvastatin	60 (20)	Significant increase of BMD at the spine and femoral hip	2 years	[148]
T2DM patients	Atorvastatin	25	No significant effect on bone turnover markers	12 weeks	[159]
Elderly women over 60 years old	Statins	3675 (928)	Decreased risk of non-pathological fracture	1 year or more	[135]
Postmenopausal women	Atorvastatin, Pravastatin, Fluvastatin, or Simvastatin	141 (41)	Improved BMD at the spine and hip	9 to 78 months	[147]
T2DM patients	Lovastatin, Pravastatin, or Simvastatin	69 (36)	Improved BMD at femoral neck, ward triangles, trochanter and total hip	15 months	[154]

* Observation without any statistical significance.

## Data Availability

Not applicable.

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
