# Peer review of "Regulation of Osteoclast Differentiation and Activity by Lipid Metabolism"

_cells, 2021, doi:10.3390/cells10010089_

Round 1

Reviewer 1 Report

Increasing evidence indicates a close correlation between dysregulated lipid metabolism and many bone diseases.  This review article summarized current advances in the role of lipid metabolism in regulating osteoclast differentiation and activity.  Emphasis was placed on the effect of cholesterol and different fatty acid species on osteoclast differentiation and function.  Recent research on how dyslipidemia affects bone metabolism, as well as the effect of statin, a lipid-lowing drug on bone health, were also reviewed and discussed.  A comprehensive mechanistic understanding of how lipid metabolism regulates osteoclast differentiation and function will be helpful for understanding bone loss in lipid-altering conditions, thereby revealing new therapeutic options applicable to pathological bone loss.

Several Concerns:

  1. The rationale of having a separate section on “Regulation by SREBPs”, while the role of SREBPs on osteoclast differentiation was included under “The role of cholesterol in osteoclasts” is not clear.
  2. It would be more reasonable to discuss “dyslipidemia and bone metabolism” first, then discuss individual pathways (cholesterol, fatty acid) on osteoclast differentiation and function.
  3. The section 10 “Statins” was piled with results from different studies without a very clear structure and logic.
  4. Several places in the manuscript mentioned that lipid metabolism regulates the function and phenotype of osteoclasts. It is not clear what the phenotype refers to, and no discussion related to how lipid regulates the phenotype of osteoclasts was seen.

Author Response

  1. The rationale of having a separate section on “Regulation by SREBPs”, while the role of SREBPs on osteoclast differentiation was included under “The role of cholesterol in osteoclasts” is not clear.

We thank the reviewer for this point. We have now moved the paragraph describing the role of SREBPs on osteoclast differentiation to the section “Regulation by SREBPs” on pp.3-4

  1. It would be more reasonable to discuss “dyslipidemia and bone metabolism” first, then discuss individual pathways (cholesterol, fatty acid) on osteoclast differentiation and function.

We thank the reviewer for this point. Although the focus of this review is on how lipid metabolism regulates osteoclast differentiation and function, it is hard to find reports clearly elucidating the association between dyslipidemia and osteoclasts. However, we believe that the topic entitled “dyslipidemia and bone metabolism” is important. Therefore, we decided to leave the section in the end.

  1. The section 10 “Statins” was piled with results from different studies without a very clear structure and logic.

We thank the reviewer for the constructive comment. We divided section10 into two parts- 10.1. Preclinical studies and 10.2. Clinical studies on pp. 10-11.

  1. Several places in the manuscript mentioned that lipid metabolism regulates the function and phenotype of osteoclasts. It is not clear what the phenotype refers to, and no discussion related to how lipid regulates the phenotype of osteoclasts was seen.

We apologize that we didn’t clearly present the description. There are several aspects of osteoclasts including differentiation, fusion, bone resorption (activity or function), and apoptosis/survival.

Reviewer 2 Report

Thank you for the opportunity to review the manuscript submitted to Cells.

In general, the authors refer to previous studies in the present tense. It is usual to refer in the past tense, such as XXX showed YYY etc. The manuscript needs to be corrected for these.

The section on lipid metabolism is well written including the regulation.

Line 131 statin(s)?

Line 162; what is meant by macrophage uptake?

Line 172: what is meant by reduced osteoclasts? Less osteoclastogenesis or activity?

Line 203: what is meant by osteoporosis mediated bone loss?

Should carnitine be included in figure 3?

Line 249: what is meant by malonyl CoA is also determined…? The sentence does not make sense.

Line 273: LCPUFA are not essential fatty acids; just LA and ALA. Do mention the minimal metabolism of ALA to EPA and DHA and why. What would be the main sources then for EPA and DHA?

Line 292: please expand on “the effect of LCPUFA on overall bone health is still inconclusive”. What is meant by bone health?

Lines 309-325: Sections are contradictory. In one sentence the authors claim that palmitic acid increases osteoclast survival and further on they claim that palmitic acid inhibits osteoclastogenesis etc. Please clarify.

Lines 326-332 can be shifted to earlier in this section as it relates to the LCPUFAs.

The whole of section 8 can be reordered to start with in vitro, followed by animal followed by human studies.

Line 349: Statement “hypercholesterolemia induced bone loss”. Reference.

Lines 364 onwards: This section can be updated to include new research showing that lean mass is associated with bone density and not fat mass.

Line 393: reference to a reference 2?

Lines 395 onwards: Statement: the positive effect of statins on bone health is mostly due to the effect of lowering lipid levels”. How?

This section under 10 can also be reordered.

Table 1: please include the time period of supplementation.

Lines 452 onwards: how could in vivo mouse models be used to test the effect of lipid synthesis on osteoclast differentiation?

Author Response

  1. In general, the authors refer to previous studies in the present tense. It is usual to refer in the past tense, such as XXX showed YYY etc. The manuscript needs to be corrected for these.

 We thank the reviewer for this point. We have made changes accordingly.

  1. The section on lipid metabolism is well written including the regulation.

We thank the reviewer for the positive feedback.

  1. Line 131 statin(s)? We have made changes in line 143 on pg.5.

  1. Line 162; what is meant by macrophage uptake?

We apologize that we didn’t clearly present the description. It means cholesterol uptake. We have changed the sentence in lines 163-167 in the revised manuscript.

  1. Line 172: what is meant by reduced osteoclasts? Less osteoclastogenesis or activity?

We have added “number of” in the text, in line 186, in the revised manuscript.

  1. Line 203: what is meant by osteoporosis mediated bone loss?

We have changed “osteoporosis mediated bone loss” to “ovariectomy-induced and age-induced bone loss.” in lines 204-205 in the revised manuscript.

  1. Should carnitine be included in figure 3?

We thank the reviewer for the insightful comment and agree with the reviewer that carnitine is an important component in lipid metabolism. However, to the best of our knowledge, carnitine and osteoclasts have not been studied. Since our review focuses on osteoclasts, we chose not to add carnitine in Figure 3.

  1. Line 249: what is meant by malonyl CoA is also determined…? The sentence does not make sense.

We have changed the sentence to “malonyl CoA can inhibit the activity of carnitine palmitoyl transferases (CPTs)” in line 251 in the revised manuscript.

  1. Line 273: LCPUFA are not essential fatty acids; just LA and ALA. Do mention the minimal metabolism of ALA to EPA and DHA and why. What would be the main sources then for EPA and DHA?

We apologize that we provided misinformation. We have deleted “essential” from the sentence.

This point is described in lines 275-276 and 279-286 on pg. 8.

  1. Line 292: please expand on “the effect of LCPUFA on overall bone health is still inconclusive”. What is meant by bone health?

We meant maintaining healthy bone. We have modified the sentence in lines 301-302 on pg. 8.

  1. Lines 309-325: Sections are contradictory. In one sentence the authors claim that palmitic acid increases osteoclast survival and further on they claim that palmitic acid inhibits osteoclastogenesis etc. Please clarify.

We thank the reviewer for this point. As we mentioned, there are contradictory studies reporting either the promoting effect or the suppressive effect of SFAs including palmitic acids on osteoclastogenesis. However, most studies demonstrated the positive effect of SFAs on osteoclast differentiation. This point is described in lines 322-323 on pg.9.

  1. Lines 326-332 can be shifted to earlier in this section as it relates to the LCPUFAs.

We moved the section at the end of LCPUFA section in lines 312-318 on pp.8-9.

  1. The whole of section 8 can be reordered to start with in vitro, followed by animal followed by human studies.

We thank the reviewer for the constructive comment. After long consideration, we decided to leave the order by FA species.

  1. Line 349: Statement “hypercholesterolemia induced bone loss”. Reference.

Reference is added as ref # 106 in revised manuscript in line 356 on pg. 9.

  1. Lines 364 onwards: This section can be updated to include new research showing that lean mass is associated with bone density and not fat mass.

We thank the reviewer for the constructive comment. The section has been updated in lines 377-382 on pg. 10.

  1. Line 393: reference to a reference 2?

We apologize; that was a typo. The sentence has been deleted.

  1. Lines 395 onwards: Statement: the positive effect of statins on bone health is mostly due to the effect of lowering lipid levels”. How?

We reached the conclusion based on the study (#138 in the revised manuscript) describing that low density lipoprotein-cholesterol lowering effect by statin mediated the improvement in bone mineral density after performing univariate and multivariable Mendelian randomization analyses.  

Zheng, J.; Brion, M.J.; Kemp, J.P.; Warrington, N.M.; Borges, M.C.; Hemani, G.; Richardson, T.G.; Rasheed, H.; Qiao, Z.; Haycock, P., et al. The Effect of Plasma Lipids and Lipid-Lowering Interventions on Bone Mineral Density: A Mendelian Randomization Study. Journal of bone and mineral research : the official journal of the American Society for Bone and Mineral Research 2020, 10.1002/jbmr.3989, doi:10.1002/jbmr.3989.

  1. This section under 10 can also be reordered.

We thank the reviewer for the constructive comment. We reorganized section 10. Please see the response to the point 3 of the Reviewer 1.

  1. Table 1: please include the time period of supplementation.

We thank the reviewer for the constructive comment. We added the time period of supplementation to the table.

  1. Lines 452 onwards: how could in vivo mouse models be used to test the effect of lipid synthesis on osteoclast differentiation?

We thank the reviewer for the constructive comment. The models to this point have been added in lines 473-475 on pg. 13.

Reviewer 3 Report

This review is about the effect of lipid metabolism on osteoclast activiy and it is well-arranged and well-written.

The present form is acceptable for publication.

Author Response

We greatly appreciate your time and effort for reviewing our manuscript. Thank you very much.

Round 2

Reviewer 2 Report

No further comments.